# Eco-Efficient Systems Based on Nanocarriers for the Controlled Release of Fertilizers and Pesticides: Toward Smart Agriculture

**DOI:** 10.3390/nano13131978

**Published:** 2023-06-29

**Authors:** Paola Fincheira, Nicolas Hoffmann, Gonzalo Tortella, Antonieta Ruiz, Pablo Cornejo, María Cristina Diez, Amedea B. Seabra, Adalberto Benavides-Mendoza, Olga Rubilar

**Affiliations:** 1Centro de Excelencia en Investigación Biotecnológica Aplicada al Medio Ambiente (CIBAMA), Facultad de Ingeniería y Ciencias, Universidad de La Frontera, Av. Francisco Salazar 01145, Temuco 4811230, Chile; n.hoffmann01@ufromail.cl (N.H.); gonzalo.tortella@ufrontera.cl (G.T.); cristina.diez@ufrontera.cl (M.C.D.); olga.rubilar@ufrontera.cl (O.R.); 2Programa de Doctorado en Ciencias en Recursos Naturales, Facultad de Ingeniería y Ciencias, Universidad de La Frontera, Av. Francisco Salazar 01145, Casilla 54-D, Temuco 4811230, Chile; 3Departamento de Ingeniería Química, Universidad de La Frontera, Av. Francisco Salazar 01145, Casilla 54-D, Temuco 4811230, Chile; 4Departamento de Ciencias Químicas y Recursos Naturales, Universidad de La Frontera, Av. Francisco Salazar 01145, Casilla 54-D, Temuco 4811230, Chile; maria.ruiz@ufrontera.cl; 5Escuela de Agronomía, Facultad de Ciencias Agronómicas y de los Alimentos, Pontificia Universidad Católica de Valparaíso, Calle San Francisco s/n, La Palma, Quillota 2260000, Chile; pablo.cornejo@pucv.cl; 6Center for Natural and Human Sciences, Universidade Federal do ABC, Santo André 09210-580, SP, Brazil; amedea.seabra@ufabc.edu.br; 7Departamento de Horticultura, Universidad Autónoma Agraria Antonio Narro, Saltillo 25315, Mexico; abenmen@gmail.com

**Keywords:** controlled release system, agrochemical, nanocarriers, crop improvement

## Abstract

The excessive application of pesticides and fertilizers has generated losses in biological diversity, environmental pollution, and harmful effects on human health. Under this context, nanotechnology constitutes an innovative tool to alleviate these problems. Notably, applying nanocarriers as controlled release systems (CRSs) for agrochemicals can overcome the limitations of conventional products. A CRS for agrochemicals is an eco-friendly strategy for the ecosystem and human health. Nanopesticides based on synthetic and natural polymers, nanoemulsions, lipid nanoparticles, and nanofibers reduce phytopathogens and plant diseases. Nanoproducts designed with an environmentally responsive, controlled release offer great potential to create formulations that respond to specific environmental stimuli. The formulation of nanofertilizers is focused on enhancing the action of nutrients and growth stimulators, which show an improved nutrient release with site-specific action using nanohydroxyapatite, nanoclays, chitosan nanoparticles, mesoporous silica nanoparticles, and amorphous calcium phosphate. However, despite the noticeable results for nanopesticides and nanofertilizers, research still needs to be improved. Here, we review the relevant antecedents in this topic and discuss limitations and future challenges.

## 1. Introduction

The constantly growing human population and the need to improve agricultural production have led to the intensive application of agrochemicals [1]. Until now, several million tons of agrochemicals are applied yearly to food crops to increase plant nutrition and reduce the attack of pathogens. Phytopathogens cause significant damage to vegetables and fruits, reaching billions of dollars per year in losses [2]. It was reported that about 22,000 species including plant pathogens, insects, concomitantly weeds, and mites attack crops globally [3]. The global consumption of pesticides reaches 2 million tons per year for controlling plant pests to ensure crop performance. Nevertheless, the inappropriate and excessive use of pesticides has enhanced the presence of hazardous residues in the environment, producing adverse impacts on natural ecosystems and humans [4]. Consequently, worldwide policies have implemented approaches and restrictions on the use of pesticides. However, mineral fertilizers based on nitrogen (N), phosphorus (P), potassium (K), and some micronutrients are widely applied by farmers to enhance food crop yield [5]. The constant and excessive application of mineral fertilizers has significantly threatened soil health, and leaching losses are associated with adverse environmental impacts [6].

The literature demonstrates that agrochemicals harm human health at neurological, respiratory, reproductive, gastrointestinal, dermatological, and endocrine levels [7,8]. Additionally, they can cause damage to animals and humans via inhalation, skin absorption, and dietary intake [9,10]. Concurrently, climate change is considered a severe problem for agricultural performance because it enhances the effects of abiotic and biotic stresses, which harm agrarian systems [11]. Climate change results in the alteration of temperature, annual rainfall, global shifts in CO_2_, ozone, and the modification of pests and microbes. In this sense, some alternatives have been focused on developing environmentally friendly products for agriculture, such as microbial inoculants and organic amendments [12]. Nevertheless, they can take time to reach the objective and adequate effects to mitigate the current problems in agriculture. Thus, the appropriate dose of agrochemicals needs to be determined to maintain a minimal ecological impact and reasonable agricultural practices, reinforcing the search for innovative technologies to meet the future demands of agriculture [13]. 

In this context, nanotechnology constitutes a revolutionary tool for smart agriculture to mitigate the harmful impact of agrochemical products and the negative impact of global climate change [14]. Promising opportunities have been identified for nanotechnology to enhance sustainable agri-food systems by maximizing agricultural outputs and minimizing inputs [15]. Specifically, nanomaterials refer to particles or assemblies of at least one dimension with a length in the 1–100 nm range [1]. Particularly, nanoparticles (NPs) used as nanocarriers can reach 1000 nm in its three dimensions. The properties of NPs allow the development of new technologies for the targeted, controlled release of agrochemicals [13]. In this sense, nanotechnology contributes to the advancement of precision farming and the targeted/controlled delivery inputs to improve productivity, efficiency, and cost-benefit.

## 2. A Smart Agricultural Technology Based on Controlled Release Systems

The problems and limitations associated with conventional agrochemicals have prompted new scientific research focusing on controlled release (Figure 1). Controlled release systems (CRSs) are applied to specifically “target organisms” using innovative technology to reduce the demand for agrochemicals in agricultural systems, implying less environmental impact. A CRS allows for the efficient and slow release of active ingredients (AIs) more actively in plants and soil, reducing human exposure and ecosystem alteration [16]. Furthermore, a CRS provides a wide variety of benefits such as durability, low toxicity, and effectiveness, which allows technological development to promote sustainable agriculture [17,18]. It was evidenced that the nanoencapsulation of AIs based on organic matrixes enhances stability, dispersibility, and solubility, resulting in the minimization of an applied dose and the possibility of encapsulating hydrophobic compounds, which are their principal advantages [19]. This modern system provides a controlled dissolution of compounds through time, reducing the environmental degradation of AIs and even increasing the solubility of those hydrophilic compounds. Consequently, a CRS allows for reducing leaching losses, soil degradation, phytotoxicity, and volatilization of agrochemicals [18].The formulation of nanoagrochemicals allows for overcoming environmental obstacles such as pH, wind, temperature, rain, and UV radiation, among others, that hinder the efficient effect of conventional agrochemicals [20,21]. The benefits of nanoagrochemicals include higher crop protection, increased nutrient efficiency, and soil fertility, among others [22]. Therefore, nanoagrochemicals are expected to be more powerful for agricultural production than their conventional counterparts, with an estimated median gain of 20 to 30% regarding crop production [23].

## 3. Nanopesticides

Synthetic pesticides are characterized by their hydrophobic nature, which leads to using organic solvents during the formulation procedure [20]. Therefore, the major problem of pesticides is associated with their poor solubility in water and ineffective action after spraying, resulting in their accumulation over years in agricultural systems [24]. Furthermore, solid pesticides have an unsatisfactory efficiency due to their large size, low solubility, and poor adhesion properties. In this sense, incorporating nanotechnology to produce nanopesticides with controlled release kinetics and enhanced permeability, stability, and solubility constitutes a vital method that can be applied in different agricultural systems, which is an important attribute supporting its massive application [25]. Nanotechnology tools extend the half-life of AIs under environmental conditions, enhancing their dispersal range and wettability [16]. Interestingly, nanopesticides are characterized by their excellent thermal stability, biodegradable nature after successful delivery, and increased target affinity. In this respect, improved target affinity allows for reducing the dose of pesticide application and optimizing their effect to control pathogens in vitro, greenhouse, and field conditions. The smaller size and higher surface area of nanopesticides improve their deposition and prolong their retention on the surface of the target, resulting in a more prolonged release period and better cost-efficiency relation [26,27]. The sustained release of pesticides is carried out with capsule erosion, passive diffusion, and osmotic-driven permeation, where the interaction between the polymeric matrix and pesticide are essential parameters to reach high loading capacity and optimal release [20,28]. In this regard, a range of nanoformulations of pesticides has been developed, such as polymeric NPs, solid lipid NPs, nanogels, nanofibers, nanoemulsions, and nanomicelles (Figure 2; Table 1).

### 3.1. Polymeric Nanoparticles

Polymeric nanospheres and nanocapsules constitute an innovative system for the slow and controlled release of pesticides with a protective, biodegradable, and eco-friendly nature [16]. These NPs do not produce harmful by-products and significantly reduce pesticide consumption with their biodegradable nature [27]. Various synthetic and natural polymers are used for the encapsulation of AIs.

Poly-ε-caprolactone has proven to be an excellent polymer to formulate nanocapsules for the controlled release of atrazine, using *Brassica juncea* as a model plant [32]. It was evidenced that nanocapsule loading of atrazine at 1 mg mL^−1^ produces severe symptoms in *B. juncea*, allowing for a decrease in the doses applied. In addition, nanocapsules of poly-ε-caprolactone containing atrazine at 2000 g ha^−1^ had an efficient effect against *Amaranthus viridis* and *Bidens pilosa*, reducing the photosystem II activity by at least 50% compared to commercial products [36]. These results suggest that the same encapsulation system is efficient at encapsulating different doses for different weed targets. Moreover, poly-ε-caprolactone containing atrazine at 3.2 × 10^12^ particles mL^−1^ (0.03 to 0.05%) had efficient activity against *Caenorhabditis elegans* by decreasing the length of worms and increasing the lethality [49]. Ref. [38] reported that nanocapsules of poly-ε-caprolactone loaded with essential oil of *Zanthoxylum rhoifolium* reduced the number of eggs and nymphs of *Bemisia tabaci*. In general, poly-ε-caprolactone is a polymer used widely to encapsulate AIs to mitigate phytopathogens. Otherwise, methoxy polyethylene glycol-poly-L-glutamic acid (mPEG-PLGA) nanoparticles containing prochloraz showed a great germicidal ability (10–90%) to reduce *Fusarium graminearum* [35]. Likewise, mPEG-PLGA nanoparticles proved to be an effective nanocarrier for the encapsulation of metolachlor, a hydrophobic pesticide [34]. In this sense, the results showed enhanced solubilization of nanoencapsulated metolachlor and more significant activity using model plants compared to the commercial formulation.

Zein NPs loaded with geraniol and *R*-citronellal showed high encapsulation efficiency (>90%), stability, and protection against UV radiation. This nanoformulation showed efficient activity against the *Tetranychus urticae* Koch mite [37]. Similarly, ref. [50] demonstrated that zein nanocapsules containing mixtures of geraniol, eugenol, and cinnamaldehyde had good activity against *T. urticae* and *Chrysodeixis includes*. Interestingly, chitosan (CHT) constitutes a natural biopolymer used to formulate nanocarriers with low toxicity due to their biodegradable nature [51]. It was demonstrated that the essential oil of the pepper tree encapsulated in CHT NPs decreased the number of viable spores of *Aspergillus parasiticus* by 40–50% [29]. Moreover, oleoyl-CHT NPs at 2 mg mL^−1^ decreased the mycelium growth of *Verticillium dahlia* by altering hyphae, cytoplasm, and cell wall morphology [52]. Functionalized CHT NPs with β-cyclodextrin containing carvacrol or linalool have acaricidal and oviposition activities against *T. urticae* because of the massive vacuolation of the cytoplasm, cell wall plasmalemma separation, and missing membranous organelles [30]. Furthermore, using lanthanum-modified, CHT-oligosaccharide NPs to load avermectin demonstrated the efficiency of this nanoformulation to improve protection and resistance to disease in rice [53]. Generally, CHT and modified CHT properties allow the encapsulation of AIs with different chemical natures. Lignin is another natural polymer used to formulate nanocarriers for the controlled release of pesticides; for example, *Brassica rapa* plants exposed to diuron-loaded lignin NPs presented enhanced chlorosis symptoms and mortality compared to commercial diuron [54]. Moreover, pyraclostrobin-loaded, lignin-modified nanocapsules demonstrated significant protection of tomato plants against *Fusarium oxysporum* f. sp. *radicis*-*lycopersici*, even reducing pesticide residues in the soil [33].

### 3.2. Nanoemulsions

Nanoemulsions constitute a colloidal system with particles ranging from 20 to 500 nm, composed of organic and water phases [55]. Pesticides are formulated principally using oil-in-water nanoemulsions due to the improved dissolution of hydrophobic compounds [56]. These properties enhance their bioavailability and performance, reducing organic solvent use in traditional pesticide formulations [57]. Essential oils of *Ageratum conyzoides*, *Achillea fragrantissima*, and *Tagetes minuta* encapsulated into nanoemulsions increased the control of *Callosobruchus maculates*, reaching higher toxicity indices for ovicidal, adulticidal, and residual activities [44]. Furthermore, pulegone with sunflower essential oil-based nanoemulsions demonstrated an efficient activity against *Sitophilus oryzae* and *Tribolium castaneum*, providing powerful bioactivity (>90% mortality rates) [42]. Similarly, nanoemulsions based on the essential oil of *Mentha piperita* showed the control of *Aphis gossypii* (LC_50_ 3879.5 μL AI L^−1^) [43].

Furthermore, nanoemulsions containing clove and lemongrass oil at 4000 mg L^−1^ disrupted the cell membrane integrity in *F. oxysporum* f.sp. *lycopersici*, and plant assays identified a significant reduction in severity in tomato plants [45]. Similarly, neem oil nanoemulsions were identified for the control of *Aspergillus flavus* and *Penicillium citrinum* [39]. On the other hand, studies have reported the efficacy of nanoemulsions for the encapsulation and controlled release of synthetic pesticides. The potential antifungal effects of nanoemulsions containing mancozeb and eugenol against *Glomerella cingulata* were reported, reducing their toxicity in the soil environment [41]. Interesting results have even been reported with dual-functionalized nanoemulsions containing validamycin and thifluzamide for controlling *Rhizoctonia solani* due to decreased pesticide resistance [40]. In general, nanoemulsions are reported to have wide applications for encapsulating natural and synthetic pesticides with relevant biological activity, supporting their broader spectrum of applications.

### 3.3. Lipid Nanoparticles

Lipid NPs are a modern technology that offers carrier systems for the sustainable release of hydrophobic compounds, reducing their losses due to leaching and degradation [58]. These NPs have been implemented in the medical industry with positive results due to their eco-friendly properties and low toxicity, so their application in agriculture has attracted attention. Preliminary, it was evidenced that carbendazim and tebuconazole presented an effective interaction with a lipid matrix composed of glyceryl tripalmitate, reducing their toxicity and allowing for testing its application [59]. Furthermore, insecticidal effects of solid lipid NPs containing the essential oil of *Ziziphora clinopodioides* demonstrated a more significant toxicity effect against *T. castaneum* (LC_50_: 30.602 µL. L air^−1^) compared with pure essential oil (LC_50_: 68.303 µL. L air^−1^) [46].

### 3.4. Nanogels

Nanogels are cross-linked, three-dimensional polymer networks at the nanoscale, mixing the properties of hydrogels and NPs [60]. It was revealed that nanogels composed of hard segments of polyethylene glycol and 4,4-methylenediphenyl diisocyanate for the controlled release of λ-cyhalothrine improved pesticide exposure area and target contact efficiency, reducing aquatic pesticide exposure and foliar UV degradation [47]. Moreover, a nanogel suspension constructed from poly-vinyl alcohol-valine derivatives and lignosulfonate containing emamectin benzoate improved insecticidal efficacy against *Plutella xylostella*.

### 3.5. Nanofibers

Nanofibers are considered attractive one-dimensional nanostructures due to their high surface area, porosity, and safety compared to other nanomaterials. Nanofibers constitute a specific tool for pheromone encapsulation due to their high porosity and surface area. Non-woven nanofibers containing cypermethrin, (*E*)-8,(*Z*)-8-dodecenyl acetate, and (*Z*)-8-dodecanol demonstrated a significant effect on the control of *Grapholita molesta*, where only pheromone and pheromone mixed with insecticide triggered mortality in tarsal contact assays (>87%) [48]. Similarly, nanofibers composed of polyhydroxybutyrate, cellulose acetate, and polycaprolactone containing 1,7-dioxaspiro[5.5]undecane and (Z)-7-tetradecenal released by *Bactrocera oleae* and *Prays oleae* were nearly twice more effective at attracting both species compared with the non-encapsulated compounds [61]. These results demonstrated that encapsulating pheromones in nanofibers constitutes an excellent alternative to developing sustainable strategies to control insects or other organisms sensitive to these compounds. Additionally, the coating of soybean seeds with electrospun cellulose diacetate nanofibers containing fluopyram or abamectin showed no harmful effects on seed germination and high antifungal activity against *Alternaria lineariae* (~50%) [62].

### 3.6. Stimuli-Responsive Nano-Based Pesticides

Stimuli-responsive systems are also a potential strategy to improve the performance of controlled release properties, promoting site-specific effects and an intelligent release of pesticides in response to environmental stimuli [63,64]. In this sense, nanocarriers that respond to environmental stimuli such as temperature, pH, light, redox conditions, and enzymes play an essential role in improving the tolerance of crops to environmental stresses. Although this type of nanocarrier was developed recently, its results are promising for the control of economically important pests. Stimuli-responsive nanocarriers can be formulated with natural polymers (i.e., chitosan, starch, alginate, and cyclodextrin, among others), silica, and pillararenes, among others (Table 2). The design of nanocarriers using stimuli-responsive properties considers the inclusion of a particular active compound capable of responding to signals or modifications in the surrounding environment [65,66].

#### 3.6.1. Responsive to pH

pH-responsive polymers are used to formulate pesticide nanocarriers due to their easily controlled properties and sensitivity. Ionizable functional groups such as amines, phosphates, sulfonates, pyrimidines, and carboxylates are added to the carrier structure to establish ionic or covalent interactions. Ionizable groups are added to polymers to produce protonation or deprotonation in a specific pH medium, allowing the strengthening or weakening of electrostatic interactions that modulate the release of AIs [14].Changes in the pH of the medium trigger swelling or contraction in the nanocarriers to produce the release of AIs [14]. For example, pH-sensitive material with acid groups in its structure (i.e., –COOH, SO_3_H) swells with exposure to a basic medium. Oppositely, pH-sensitive material with basic groups (i.e., –NH_2_) swells with exposure to an acid medium.

NPs formulated with chitosan/tripolyphosphate for carrying hexaconazole reached 73% encapsulation efficiency and the fastest release at pH 4 compared with environments at pH 7 and pH 10 [67]. Moreover, a cytotoxicity assay revealed that NPs were less toxic than commercial products. Other studies demonstrated that polydopamine-modified attapulgite–calcium alginate hydrogel nanospheres promote the controlled release of chlorpyrifos in response to pH variations in the range of 5.5 to 8.5 for the control of grubs, reaching a mortality rate between 42 and 100% [68]. This evidenced the ability to respond to a specific pH stimulus, driving the development of intelligent products. Furthermore, NPs composed of poly-γ-glutamic/chitosan were an efficient system to encapsulate and release avermectin under alkaline conditions (pH 8.5) [31]. The improvement in pesticide efficiency was studied by Chen et al. [77], who showed that leaf-adhesive avermectin nanocapsules prepared with chitosan presented good thermal stability and photostability in response to UV radiation. This study demonstrated that these nanocapsules exhibited a significant release of avermectin in response to low pH. This study suggests the importance of multiparametrically analyzing the formulation of nanocarriers for environmental parameters directly related to their application, such as photo- and thermostability. Similarly, a composite nanocarrier formulated with functionalized boron nitride nanoplatelets with trimethoxysilane and poly-ethylene-glycol-diacrylate to encapsulate avermectin demonstrated a two-to three-fold increase in the release rate when changing from pH 7 to pH 11 [78]. This nanocarrier also decreased the degradation rate of avermectin from UV irradiation (30%). Both studies suggest the importance of optimizing the polymers and the factors that influence the response range to pH. It was reported that pH-responsive, polydopamine-coated graphene oxide nanocomposite is an excellent system for the release of hymexazol, with an extended persistence under a rainwash experiment, when simulated from pH 5 to 9. Additionally, its inhibition activity against *F. oxysporum* sp. *Cucumebrium* showed similar results compared to technical hymexazol [79]. In another study, a composite prepared with zeolitic imidazolate for the smart release of prochloraz at pH 5 was significantly greater than at pH 8 under light conditions and exhibited good stability under UV irradiation [70]. More recently, ref. [80] reported that responsive mesoporous silica NPs improve the controlled release of prochloraz, with higher cumulative release at pH 4 compared to pH 7 and 10.

#### 3.6.2. Responses to Temperature

Temperature-sensitive nanocarriers have great potential for application in agriculture due to the constant temperature change associated with the appearance of weeds, insects, and fungi. The formulation of nanocarriers in response to temperature stimuli is based on changes in the physicochemical properties of the polymer as the temperature shifts, which allows the release to be reversible and intelligent. Thermo-sensitive polymers are soluble in water at low temperatures, and as the temperature increases and exceeds the T° of phase transition, a polymeric separation occurs, allowing the release of AIs. The phase separation of a homogeneous polymer at high temperatures triggers AI release [14].

A core–shell structure formulated with attapulgite/NH_4_HCO_3_/amino-silicon oil/poly-vinyl alcohol was developed for the controlled release of glyphosate, where amino silicon oil and poly-vinyl alcohol formed the shell [71]. The porous structure of attapulgite can link to glyphosate, and NH_4_HCO_3_ creates nanopores to the glyphosate. The results exhibited that at 40°C, a greater release occurs compared with exposure at 25°C (~2-fold). It was also shown that after seven days, the formulation had great efficiency in the control of *Zoysia matrella*, even under a simulated rainfall pot experiment. On the other hand, a core–shell polydopamine@PNIPAm nanocomposite constitutes a good temperature-responsive system for the controlled release of imidacloprid. Indeed, a slow controlled release of imidacloprid occurred at 15 and 25 °C, while at 40 °C, a faster release was obtained over 20 h [72]. Micelle formulated with poly[2-(2-Methoxyethoxy)-ethyl- methacrylate-co-Octadecyl -methacrylate]/monomethoxy-(PEG)-13-poly(D,L-lactide-co-glycolide) and monomethoxy-(PEG)-45-poly-(D,L-lactide) was an effective nanosystem to encapsulate natural pyrethrins [81]. A nanocarrier formulated with poly-(N-isopropylacrylamide)-modified graphene oxide to control the release of lambda-cyhalothrin exhibited a persistent release with good dispersion stability. It was evidenced that there was an increase in released behavior as the temperature increased from 27 to 35°C for 7 days [82]. These systems clearly demonstrate their ability to release pesticides at room temperature (~25 °C). However, they show greater release above 35 °C, which limits their application to hot climates.

#### 3.6.3. Response to Light

Photo-sensitive nanocarriers are based on the release of AIs with light irradiation at different wavelengths (i.e., UV range, visible, and infrared). In the agricultural industry, these nanocarriers are particularly interesting due to the abundance of solar irradiation. Incorporating coumarin, spyropyrane, azobenzene, or ortho-nitrobenzyl triggers a light-activated agitator system that releases AIs with changes in the polarity or degradation of the polymer. The release of AIs from these nanocarriers is based on modifications triggered by irradiation at a specific wavelength, which produces changes in properties such as polarity, conformation, conjugation, charge, etc. [14].

Photoresponsive core–shell micelle formulated with poly(ethylene-glycol) and a photolabile *o*-nitrobenzyl group was effective in the controlled release of dichlorophenoxyacetic acid (2,4-D) [73]. Similarly, a photoresponsive system based on coumarin for the sustainable release of 2,4-D demonstrated its effectiveness with good thermal stability in crops of *Cucurbita maxima* [75]. Moreover, it was reported that carboxymethyl–chitosan conjugate with 2-nitrobenzyl side groups was synthesized, forming micelles with a core–shell configuration to encapsulate diuron, producing the controlled release of the AI with exposure to UV radiation at 365 nm [74]. This study also observed a release rate of 96.8% for diuron under solar-stimulated irradiation for 8 h. Furthermore, the photochemical properties of coumarin were studied to formulate a photoresponsive nanosystem for spirotetramat-enol controlled release [76]. The results showed that the nanosystem triggered the release of spirotetramat-enol with insecticidal effect against *Aphis craccivora* under blue light (420 nm) or sunlight conditions, thus indicating its control. Furthermore, a nanocomposite formulated with attapulgite, biochar, azobenzene, and amino silicon oil for the controlled release of glyphosate showed an excellent light-motivated system, exhibiting good herbicidal activity (~50%) against Bermuda weed leaves for 25 days [69].

These studies support the efficiency of nanocarriers in response to environmental factors. However, scaling their production and study system are key to achieving the development of a technology within the reach of all users and diverse agricultural systems.

## 4. Nanofertilizers

Agricultural food production should be increased between 60–70% for future food demand. Nevertheless, this increase in production is highly dependent on the availability of soil nutrients [83]. It was reported that 111,591, 49,096, and 40,232 thousand tons of N, P, and K, respectively, would be required by 2022 [5]. Nevertheless, it was reported that 80 % of P, 60% of K, and 50% of N are lost into the environment and are not taken up by plants. A small fraction of mineral fertilizers is incorporated into plant composition, evidencing the low nutrient use efficiency (NUE) of N (30–55%)- and P (18–20%)-based fertilizers. In this sense, nanofertilizer formulations with high efficiency are needed to reduce nutrient losses and adverse impacts in ecosystems [13].

Nanofertilizers enhance root nutrient uptake by improving soil nutrient management, relieving the nutrient resource, decreasing the immobilization of nutrients, and minimizing nutrient loss in the environment and agricultural wastes (Figure 3) [84]. The improvement in NUE using nanofertilizers could be 20–30% compared to conventional mineral fertilizers. Nanotechnology research applied in crop nutrition is expected to reach a high precision on plant targets to prevent and minimize fertilizer loss [83]. Nanoparticles have important physicochemical properties that enhance the strong attachment of fertilizers or AIs to plant surfaces by improving surface tension, and nanocoating provides high protection for larger particles [85]. The controlled release of fertilizers from nanocarriers provides nutrient longevity in the agro-environment, giving a continuous supply to crops and improving NUE [86,87]. According to [24], more than 102 nanofertilizers are currently marketed in 17 countries, where most are synthesized in Germany, China, and the USA. Thus, governments, companies, the research community, and the public sector have garnered significant interest in nanofertilizers to optimize agricultural systems. The efficient potential of nanocarriers for the controlled release of fertilizers has been revealed, supporting their capacity to replace or decrease the application of conventional mineral fertilizers (Table 3).

### 4.1. Nanohydroxyapatite

Hydroxyapatite (Ca_10_(PO_4_)_6_(OH_2_)) NPs (HA NPs) are derived from natural (i.e., bovine and horse wastes) and synthetic (i.e., chemical deposition and electrodeposition) pathways, differentiating through the presence of some ions such as CO_3_^2−^, Si^2+^, Mg^2+^, Zn^2+^, K^+^, and Na^+^ [83]. HA NPs stabilized with carboxymethylcellulose in concentrations from 200 to 2000 mg L^−1^ increased the primary root elongation of *Solanum lycopersicum* plants grown under hydroponic conditions for 2 weeks [89]. In addition, these results suggest that HA NPs constitute an efficient P supplier with a significant potential to be a carrier of nutrients. In addition, HA NPs can be an efficient tool to increase P soil mobility, which significantly increases the root and foliar biomass of plants [114]. Interestingly, the functional groups present on the surface of hydroxyapatite allow the immobilization of chemicals to generate nanohybrids. A nanohybrid suspension synthesized by coating HA with urea in a proportion 6:1 urea:HA by weight presented a slow release and increased its yield and nitrogen, phosphorus, and potassium (NPK) content in leaves [88]. Moreover, HA NPs loaded with urea enhanced the amylase and starch content, fresh and dry weights, and seedling growth of *O. sativa* [90]. Similarly, urea–HA NPs showed an increase in total polyphenols, amino acids, and brightness in leaves of *Camellia sinensis*, reducing the urea applied by 50% and increasing the yield by 10–17% [91]. An interesting study by Rop et al. [115] evidenced that a composite formulated with mineral fertilizers (urea, diammonium hydrogen phosphate, and potassium sulfate) and HA NPs into hyacinth cellulose-graft-poly-acrylamide hydrogel promotes an increase in P content from 8 to 16 weeks, while mineral N significantly increased from 8 to 12 weeks, thus demonstrating that HA NPs are a relevant alternative to synchronize nutrient release according to nutritional requirements. Furthermore, modified urea–HA NPs have shown a slow release of Ca^2+^, PO_4_^3−^, NO_2_^−^, NO_3_^−^, Fe^2+^, Zn^2+^, and Cu^2+^,resulting in enhanced nutrient uptake in *Abelmoschus esculentus* [92]. Another interesting study performed by Yoon et al. [93] reported that humic substances bond to HA NPs, improving the growth and yield of *Zea mays* and promoting tolerance to NaCl stress. The studies described above present significant evidence supporting the use of HA NPs as a tool to mobilize P and even encapsulate N sources to increase NUE. However, it is necessary to scale experimental tests at the field level to assess its massification in different agricultural systems

### 4.2. Nanoclays

Layered double hydroxides (LDHs) are composed of layered hydroxides with divalent (M^2+^) and trivalent (M^3+^) cations. LDHs have been proposed as systems to slow the release of P fertilizers and to adsorb phosphate during the recovery of P from the waste stream [94]. Mg-Al LDHs with varying M^2+^/M^3+^ ratios were synthesized as NO^3−^ were exchanged with PO_4_^2−^, where the P efficiency of P-LDH was 4.5 times higher compared to soluble P under acid soil conditions. However, the P use efficiency decreased in calcareous soil, reaching above 20% of soluble P forms relative to the total P amount. These results suggest the importance of carrying out tests with different types of soil since their physicochemical properties are crucial. Likewise, an LDH intercalated with phosphate ions was tested to evaluate P fertilization in tropical weathered soils (sandy and clayey soils), using *Z. mays* as a plant indicator [96]. The results indicated that LDH phosphate improved the productivity, P content, and height of *Z. mays* and promoted an increase in soil pH, which resulted in improved P availability.It was shown that a P content of ~40 mg g^−1^ hydrotalcite-like LDH ([Mg-Al]-LDH) released phosphate at a 10-fold slower rate compared to KH_2_PO_4_, where the interaction between P and Fe^3+^ stood out in the soil [97]. It was noted that similar assays with *T. aestivum* showed that [Mg-Al-PO_4_]-LDH generated the same level of P nutrition as other conventional sources and maintained the phosphate concentration for a long time. On the other hand, two forms of Zn-Al LDHs associated with borate showed a slow release of Zn and B for 28 days, where only monoborate ions participated in the intercalation and adsorption phenomena [98]. In addition, experiments performed using *S. lycopersicum* plants showed remarkable results with widespread application of Zn-Al LDHs and NPK fertilizer, which increased dry mass (~5-fold) and P-K-B-Zn contents (~10-fold), thus reducing the loss of soil nutrients. These results support the efficient slow release of Zn and B from LDH to reinforce the effect of mineral fertilizers containing different nutrients with agronomical importance.

In another study, nano-zeolite and zeolite-nanocomposites exhibited significant results regarding the long-term availability of macro and micronutrients such as P, Na^+^, K^+^, Zn^2+^, Ca^2+^, Mg^2+^, NO_3_^−1^, and Fe^+3^ in soil for 14 days [116]. Moreover, the findings showed that zeolite-nanocomposites increased the water retention capacity (15–20%), while nano-zeolite was advantageous in maintaining the soil water level. Furthermore, exciting results were obtained using saturated nano-zeolite with (NH_4_)_2_SO_4_ plus nano-HA and saturated nano-zeolite with (NH_4_)_2_SO_4_ plus triple phosphate, which increased the height, branch number, flower number, P content at the root and shoot levels, and the fresh and dry weight of shoots of *Matricaria chamomilla* [95]. In another study, kaolinite, illite, and smectite nanoclays were used to synthesize polymer composites loaded with di-ammonium phosphate or a urea solution, which demonstrated a considerable increase in cumulative P and total mineral N when used in Inceptisols, Alfisols, and Vertisols, potentiating their application in diverse agricultural systems to reduce nutrient losses [117].

### 4.3. Chitosan Nanoparticles

CHT is a natural polymer widely studied for the encapsulation and controlled release of active ingredients. CHT NPs emerge as an important tool for stimulating plant growth by activating physiological parameters. For example, Zn-loaded CHT-NPs demonstrated a capacity to mitigate *T. aestivum* stress under the deficiency of this micronutrient with a foliar application twice a week, which increased the Zn content in grains by 27–42%. This result suggests the potential application of nanocarriers to improve the biofortification of nutrients in crops [99]. Similarly, Zn-CHT NPs applied using seed priming and foliar in *Z. mays* showed significant results in improving plant immunity by enhancing defense enzymes and antioxidant levels, increasing grain yield, and fortifying Zn micronutrients in grains [101]. Furthermore, Cu-CHT NPs at 0.01 and 0.16% improved antioxidant activities and defense enzymes of *Z. mays*, even increasing its growth under greenhouse and field conditions [100].

Furthermore, the encapsulation of NPK nutrients into CHT NPs showed interesting results, increasing photosynthesis traits, nutrient uptake, and growth of *Coffea arabica* coffee plants [118]. Similarly, ref. [119] reported that CHT NPs provide a good release of urea, calcium phosphate, and potassium chloride in a water solution, and the nutrient loading influences the stability of CHT NPs. In particular, incorporating K into CHT NPs at 75% increased the fresh and dry biomass of *Z. mays* and improved physical properties such as porosity, water conductivity, and friability to improve root development [102]. Another related study showed that the exposure of *Z. mays* plants for 10 weeks to NPK-CHT NPs increased the height, number of leaves, stem diameter, and chlorophyll content by improving the content and uptake of nutrients [120]. Additionally, the nanoformulation increased soil microbiological and root activity, suggesting that a synchronized nutrient release from CHT NPs reduces fertilizer requirements and environmental impacts.

Moreover, results regarding the use of chitosan–urea nanocomposites showed an exciting increase of soil dehydrogenase activity and organic carbon content. Meanwhile, the slow release of urea reduced the concentration of NH_4_^+^-N and NO_3_^−^-N in a soil cropped with potatoes, influencing microorganism populations associated with the soil N cycle [103]. On the other hand, the incorporation of poly-γ-glutamic acid into CHT NPs was an efficient system to encapsulate gibberellic acid (GA_3_), which also showed a sustainable release for 48 h and significantly increased the germination rate of *Phaseolus vulgaris* seeds, as well as strongly increasing the leaf area and root development [121]. Furthermore, Leonardi et al. [104] reported the efficiency of CuO- chitosan/alginate NPs for the slow release of Cu, which also was associated with an improvement in the seed and seedlings of *Fortunella margarita*, benefiting the development of epigean.

### 4.4. Mesoporous Silica Nanoparticles

Mesoporous silica nanoparticles (MSNs) are an attractive system for formulating a controlled release system due to their properties such as porosity, versatile surface functionalization, controllable pore size, stability, biocompatibility, high surface area, and low toxicity [122]. Lupin and wheat exposed to MSN NPs at concentrations from 500 to 1000 mg L^−1^ increased plant biomass, total protein, chlorophyll content, seed germination, and photosynthetic activity, and interestingly, high concentrations (2000 mg L^−1^) did not produce oxidative stress in the plants [105]. A study performed with auxin on MSN Au/SiO_2_ NPs evidenced an increase in embryogenesis, ploidy, calli induction frequency, calli length, number of regeneration zones, and methylation levels in *Linum usitatissimum* [106]. In addition, an MSN nanocomposite of ZnAl_2_Si_10_O_24_ evidenced significant results in the simultaneous slow release of Zn and urea, providing an efficient system to fertilize *O. sativa* when used at concentrations of 60 to 150 mg kg^−1^ compared to commercial urea [107]. In general, MSNs present interesting results, but testing their effects on species of vegetables and fruits of economic interest is crucial to determine their development prospects.

### 4.5. Amorphous Calcium Phosphate

A study showed that calcium phosphate nanoparticles (CaP NPs) containing mycorrhizal (*Glomus mosseae*) and endosymbiont (*Piriformospora indica*) fungi improved the performance of *Z. mays* by enhancing the growth of roots and shoot leaves (length and weight) and chlorophyll content [108]. These results suggest a synergistic growth promotion produced by Ca and phosphate nutrients released by nanocomposites and the effect of both fungi to improve nutrient uptake. Interestingly, the foliar application of CaP NPs loaded with urea generated significant results for improving grapevine nutrition by increasing the concentration of amino acids and arginine, suggesting the ability of nanotechnology tools to reduce the N dose application in fruits and maintain the fruit quality during harvest stage [109]. Furthermore, a study with CaP NPs doped with K, nitrate, and urea showed a more controlled release profile, improving the grain yield of *Triticum durum* with a reduction of 40 wt % of the N applied with respect to conventional mineral fertilizer, suggesting that this system can be more efficient at promoting a sustainable nanofertilizer tool [111]. Moreover, Ramírez-Rodríguez et al. [110] revealed that urea-doped nanofertilizer was an effective nanocarrier system for adsorbing urea with a reduced N dose (40%), which increased the yield and quality of *T. durum* by increasing the weight, shoot number, kernel weight, and protein content under both controlled and field conditions. The uptake of nutrients by roots was faster compared to the leaves. In another study, interesting results were obtained in *Vitis vinifera* cv. Pinot Gris exposed to urea-doped CaP NPs under semi-controlled conditions, which showed that vine plants recognize and assimilate the N provided by CaP NPs [112]. Furthermore, this study demonstrated that the foliar and fertigation applications of urea-doped CaP NPs in *V. vinifera* had the same effects compared to commercial granular fertilizer in chlorophyll (SPAD index), yield, bunch weight, and amount of yeast-assimilable N. Likewise, ref. [113] reported that urea-functionalized CaP NPs are an efficient system to optimize the growth of *Cucumis sativus* using the half N content, which increases the root and shoot biomass in an equivalent amount compared to conventional fertilizer but without N losses. According to what has been described, these NPs have a high application potential due to their important results for species of economic interest.

## 5. Conclusions and Future Directions

Nanotechnology provides innovative strategies to improve agricultural productivity and find solutions to several environmental issues associated with limited water resources, soil deterioration, energy crises, and climate change [16]. Nanotech tools have demonstrated efficient results for a wide range of applications in agriculture to enhance crop production and yield [123]. Several studies evidenced the relevant role of nanoagrochemicals in improving crop yield, but research is still in the early stages [124]. The literature shows the beneficial effects of nanoagrochemicals, which depend mainly on their physicochemical properties, exposure time, target organisms, and environmental conditions [22]. Therefore, careful consideration of parameters such as dose, delivery strategy, and establishing experimental conditions are required to evaluate these products.

Nowadays, researchers are focused on developing nanocarriers with safety properties and efficient responses to the sustainable and focused release of nutrients and pesticides [25]. The development of nanocarriers is projected to optimize their effects on the target organism and to improve the sustainable release of compounds or agrochemicals, minimizing the losses derived from premature degradation, leaching, and volatilization [13,23,124]. Thus, it is expected that controlled release systems based on nanocarriers to improve the growth and protection of plants will be an essential tool to overcome the environmental issues derived from conventional agrochemicals in future agriculture [125]. Environmental safety and the potential risks of nanoagrochemicals to non-target organisms in the ecosystem and human health should be assessed to ensure adequate management [126]. Until now, risk assessments of nanoagrochemicals have focused mainly on experimental tests carried out under laboratory conditions, while the real scenarios in agricultural systems have not yet been investigated in depth. Therefore, innovative approaches should be developed to optimize the delivery, uptake, targeting, and long-term effects under field conditions to determine the effectiveness and environmental risks.

In addition, future studies should be performed to evaluate the interaction between nanoagrochemicals under natural conditions and other non-target organisms in the ecosystem. The interaction between nanoagrochemicals applied at the foliar and root levels of plants should be evaluated to determine potential risks and harmful effects in the food chain. In this sense, proteomic, genomic, and metabolomic studies can help understand the mechanisms involved in interacting with the exposed organisms [80]. A robust interdisciplinary analysis of the impact of nanoagrochemicals in ecosystems and the food chain should be performed to determine their implications for environmental fate, which is essential to establish policy decisions and market status. Currently, there is no defined compilation of safety criteria that regulatory agencies can use to approve novel nanoproducts, although the European Food Safety Authority and OECD guidance for testing soil leaching and toxicity of nanomaterials are important contributions to validate nanoagrochemicals [127,128].

Hence, the application of nanoagrochemicals in agriculture requires great efforts from various disciplines (i.e., scientific researchers and regulators) to overcome the current difficulties resulting from the lack of knowledge about the implications of the application of nanoagrochemicals in agricultural systems and their real effect on the adequate functioning of the ecosystem [124]. Several perspectives must be integrated using different viewpoints related to science, industry, regulators, and the public to critically assess progress in the application of nanoagrochemicals. Many efforts have been made in the last years to develop international management strategies to evaluate the risks and potential hazards of nanoagrochemicals [129]. In addition, innovative approaches to testing nanoagrochemicals at the laboratory scale should be designed for soils and in field studies to demonstrate their efficacy under realistic agricultural conditions. Nowadays, nanoagrochemicals offer a range of benefits, and some companies have developed protocols for their production and application, but commercial products still need to be improved further. At the retail level, the cost, complexity formulation, and high demand for qualified personnel have limited production at the industrial scale. The high cost of production and low-margin industry are major constraints to scale production [24]. Furthermore, the lack of standards and uniform methodologies for establishing regulations associated with applying nanoagrochemicals in agriculture and food has prevented adequate evaluation. Therefore, there is an urgent need to develop uniform methods to evaluate the safety risks for their long-term application under field conditions. Finally, ethical issues about the use of nanoagrochemicals must be considered due to their effects on food and agribusiness and the scarce reliable information on the application of these products under real conditions in agriculture.

## Figures and Tables

**Figure 1 nanomaterials-13-01978-f001:**
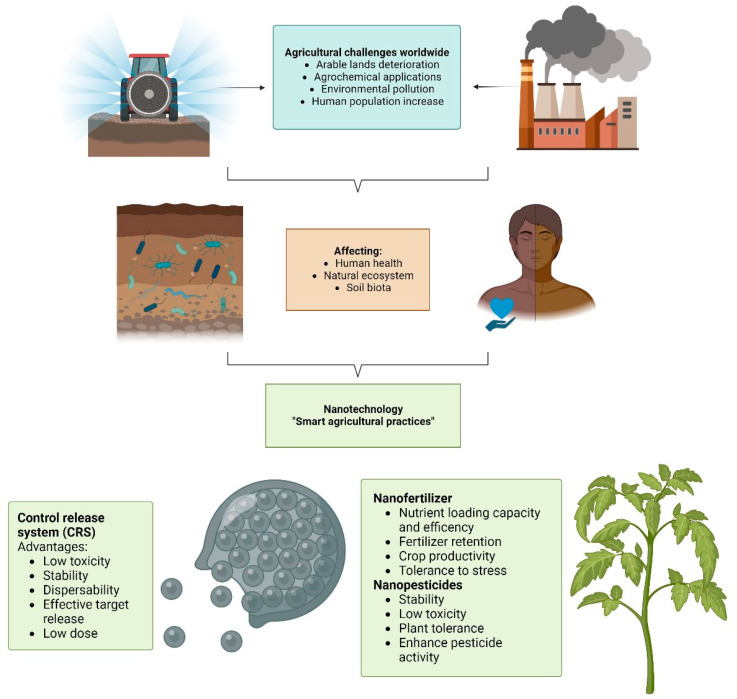
The main challenges in agriculture and the advantages of controlled release nanoproducts.

**Figure 2 nanomaterials-13-01978-f002:**
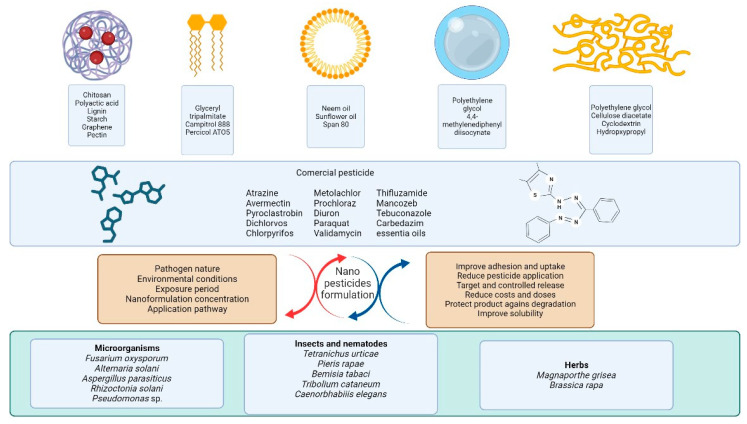
The principal nanoformulations used for the controlled release of commercial pesticide and advantages for the control of phytopathogens.

**Figure 3 nanomaterials-13-01978-f003:**
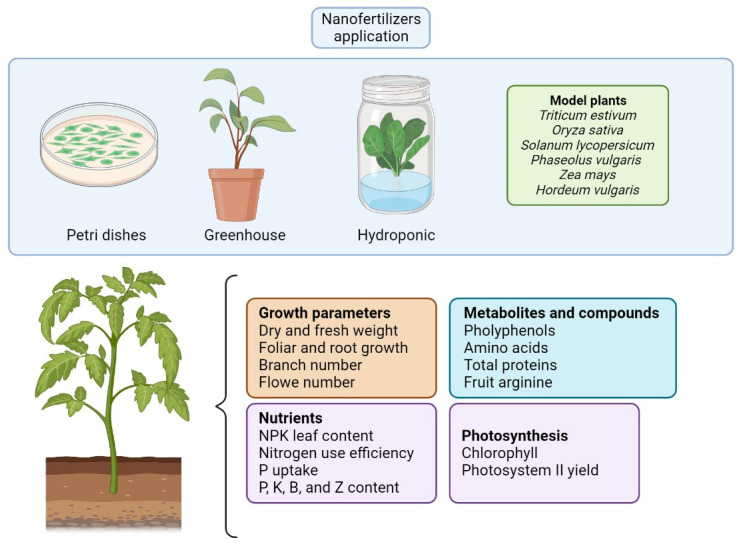
The principal nanostructures for the controlled release of fertilizers and their effects on model plants.

**Table 1 nanomaterials-13-01978-t001:** Summary of different types of potential nanopesticides based on controlled release systems.

Formulation	Active Ingredient	Size (nm)	Target Organism	Suppresion Effect	Compared to Control	Reference
Nanocapsules						
Chitosan	Pepper tree essential oil	20–100	*Aspergillus parasiticus*	Viability	40–50%	[29]
Chitosan funcionalized with β-cyclodextrin	Carvacrol linalool	175.2–245.8	*Tetranychus urticae*	Repellency	>80%	[30]
Chitosan	Avermectin	310	*Magnaporthe grisea*	Blast fungus	2-fold	[31]
Poly(ε-caprolactone)	Atrazine	240.7	*Brassica juncea*	Dry weight	10-fold	[32]
Lignin	Pyraclostrobin	162.4	*Fusarium oxysporum* f.sp. *radicis*-*lycopersici*	EC_50_	3.8-fold	[33]
mPpeg-PLGA	Metolachlor	90.49–128.7	*Oryza sativa-Digitaria sanguinalis*	Seedling height	~5.5-fold	[34]
				Root length	~10-fold	
mPEG-PLGA	Prochloraz	190.7	*Fusarium graminearum*	Germicidal efficacy	7.7%	[35]
Poly(ε-caprolactone)	Atrazine	260	*Bidens pilosa* *Amaranthus viridis*	Inhibitory growth	10-fold	[36]
Zein	Essential oil of citronella	142.5–172.3	*Tetranychus urticae* Koch mite	Repellency	200%	[37]
PCL	Essential oil of *Zanthoxylum rhoifolium*	500	*Bemisia tabaci*	Number of eggs and nymphs	95%	[38]
Nanoemulsions						
Neem oil	*Azadirachta* *indica*	59	*Aspergillus flavus* *Penicillium citrinum*	Growth inhibition	71.4%	[39]
Polylactide	Validamycin Thifluzamide	260	*Rhizoctonia solani*	Toxicity	4.2-times	[40]
Span 80	Mancozeb Eugenol	200–300	*Glomerella cingulata*	Number of juveniles	1-fold	[41]
Sunflower oil	R-(+)-pulgone	131–558	*Sitophilus oryzae* L. *Tribolium castaneum*	Mortality rates	>90%	[42]
*Mentha piperita* oil and Tween 80	*Mentha piperita* essential oil	20–60	Cotton aphid	Contact toxicity	LC_50_: ~3879	[43]
-	Essential oil of *Ageratum conyzoides*, *Achillea fragrantissima* and *Tagetes minuta*	48.6–136.3	*Callosobruchus maculatus*	Egg toxicity	LC_50_:16.1–40.5 µL L^−1^	[44]
Propylene glycol	Clove and lemongrass oil	76.73	*Fusarium oxysporum* f.sp. *lycopersici*	Severity	70.6%	[45]
Lipid nanoparticles						
Percirol ATO5 + campritol 888	Essential oil of *Ziziphora clinopodioides* Lam.	241.1	*Tribolium castaneum*	Mortality	100%	[46]
Nanogels						
Polyethylene glycol 4,4-Methylenediphenyl diisocyanate	λ-cyhalothrine	120	*Athetis dissimilis*	Mortality	~60%	[47]
Nanofibers						
Poly-ε-caprolactone Polyethylene glycol	Cypermethrin (Z)-8-Dodecenyl acetate (Z)-8-Dodecanol	-	*Grapholita molesta* (Lepidoptera: Tortricidae)	Mortality	>87%	[48]

Abbreviations: FTIR: Fourier transform infrared spectroscopy, DLS: dynamic light scattering, XRD: X-ray diffraction, SEM: scanning electron microscopy, TEM: transmission electron microscopy, XPS: X-ray photoelectron spectroscopy, NTA: nanoparticle tracking analysis, TGA: thermal gravimetric analysis, SEM-EDX: scanning electron microscopy–energy dispersive X-ray spectroscopy, DSC: differential scanning calorimetry, AFM: Atomic force microscopy, NMR: nuclear magnetic resonance.

**Table 2 nanomaterials-13-01978-t002:** Summary of the main environmentally responsive controlled release systems to nanopesticides.

Response to	Polymer	AI	Condition Release	Size (nm)	Organism Target	Suppression Effect	Compared to Control	Reference
pH								
	Chitosan/tripolyphosphate	Hexaconazole	pH 4	100	*Rhizoctonia solani*			[67]
	Polydopamine-modified attapulgite- calcium alginate hydrogel nanosphere	Chlorpyrifos	pH 5.5–8.5	20	Grubs	Mortality	42–100%	[68]
	Poly-γ-glutamic acid/chitosan	Avermectin	pH 8.5	56–61	Pine wood nematode	Blast fungus	2-fold	[31]
	Chitosan	Avermectin	Low pH	251.5–258.5	Aphids	Toxicity	LC_50_: 8.1 mg L^−1^	[69]
	Zeoliticimidazolate (2-methylimidazole/2,4-dinitrobenzaldehyde/Zn(NO_3_)_2_·6H_2_O	Prochloraz	pH 5	129.6	*Sclerotinia sclerotiorum*	Antifungal effectivity	70.8%	[70]
	Bimodal mesoporous silica modified with a silane coupling agent	Prochloraz	pH 5	546.4	*Rhizoctonia solani*	Inhibition rate	80%	[21]
Temperature								
	Attapulgite/NH_4_HCO_3_/ amino silicon oil/ poly(vinyl alcohol)	Glyphosate	40 °C		*Zoysia matrella*	Control efficiency	~70%	[71]
	Polydopamine/PNIPAm	Imidacloprid	15–40°C	~250	-	-	-	[72]
	Poly[2-(2-Methoxyethoxy) ethyl methacrylate-co-Octadecyl methacrylate] /monomethoxy (polyethylene glycol) 13 -poly(D, L-l actide-co-glycolide) and monomethoxy (polyethylene glycol) 45 -poly(D, L-Lactide)	Pyrethrins	26 °C	60–120	*Culex pipienspallens* *Aedes albopictus*	Toxicity	LC_50_: 0.06–0.12 µg a.i mL^−1^	[3]
Light								
	Poly(ethylene glycol)/photolabile *o*-nitrobenzyl	Dichlorophenoxyacetic acid	After 365 nm UV light	40	-	-	-	[73]
	Carboxymethyl chitosan/photolabile 2-nitrobenzyl side groups	Diuron	365 nm UV light	140	-	-	-	[74]
	Coumarin	2,4-D	UV light		*Cucurbita maxima*	Root length	25–50%	[75])
	Coumarin	Spirotetramat-enol	Blue light (420 nm) irradiation or sunlight		*Aphis craccivora* Koch	Toxicity	LC_50_:0.08–0.11 mmolL^−1^	[76]
	Attapulgite/biochar/azobenzene/amino silicon oil	Glyphosate	UV–Vis light (365 and 435 nm)	0.5–1 μm	Bermuda weeds	Control efficiency	~90%	[69]

Abbreviations: FTIR: Fourier transform infrared spectroscopy, DLS: dynamic light scattering, HRTEM: high resolution transmission electron microscopy, XRD: X-ray diffraction, SEM: scanning electron microscopy, TEM: transmission electron microscopy, XPS: X-ray photoelectron spectroscopy, TGA: thermal gravimetric analysis, SEM-EDX: scanning electron microscopy–energy dispersive X-ray spectroscopy, DSC: differential scanning calorimetry, AFM: atomic force microscopy, NMR: nuclear magnetic resonance, BET: Brunauer–Emmett–Teller.

**Table 3 nanomaterials-13-01978-t003:** Principal nanostructures reported for formulating controlled release systems to improve nutrient efficiency.

Nanocarrier Nature	Fertilizer	Size (nm)	Plant	Exposure Period	Condition	Effect	Compared to Control	Reference
Hydroxyapatite								
	Urea	15–20	*Oryza sativa*	4 weeks	Field	Yield	~41.8%	[88]
						NK leaf content	5.9–10.9%	
	-	35–45	*Solanum lycopersicum*	2 weeks	Hydroponic (controlled conditions)	Root elongation	100%	[89]
	Urea	40–60	*Oryza sativa*	5 days	Petri dishes (controlled conditions)	Amilase content	~153%	[90]
						Starch content	~100%	
	Urea	-	*Camellia sinensis*		Field	Yield increase	10–17%	[91]
	Urea NPs of Cu, Fe, and Zn	39.76	*Abelmoschus* *esculentus*	14 days	Field	Fe nutrient uptake	~2-fold	[92]
	P	75–125	*Zea mays*	3 months	Pot experiment (controlled conditions)	Dry weight/unit P	~100%	[93]
						Corn grain productivity	~35%	
						Resistance to NaCl stress (dry weight/unit P)	~300%	
Nanoclays	Phosphate	20	*Hordeum vulgare*	17 days	Pot experiment	P efficiency	4.5-times	[94]
	Satured nano-zeolite with (NH_4_)_2_SO_4_ plus nano-HA and satured nano-zeolite with (NH_4_)_2_SO_4_ plus triple phosphate	<100	*Matricaria chamomilla*	-	Greenhouse experiment	Height	72.5%	[95]
						Branch number	168.4%	
						Flower number	292.9%	
						Phosphorus content	85.7%	
						Fresh weight	~180%	
						Dry weight	~100%	
	Phosphate	-	*Zea mays*	25 days after sowing	Growth chamber	Dry matter	~11.5%	[96]
						P content	~29%	
						Height	~7.1%	
						Soil pH	~18%	
	Phosphate	-	*Triticum aestivum*	30 days	Pot experiment	Dry matter	122.2%	[97]
						Phosphate content	~10.3-fold	
						Available phosphate	~24.6-fold	
	Zinc, boro	-	*Solanum lycopersicum*	2 weeks	Pot experiment	Dry mass	~6–10-fold	[98]
						P content	~10–16-fold	
						K content	~13–18-fold	
						B content	~9–16-fold	
						Zn content	~8–10-fold	
Chitosan								
	Zn	250–300	Wheat	5 weeks	Pot experiment	Zn content	27–42%	[99]
	Cu (0.01%)	361.3	*Zea mays*	95 days	Field	Height	7.8%	[100]
						Ear length	15.3%	
	Zn (0.01%)	200–300	*Zea mays*	95 days	Pot experiment	Grain yield	19.3%	[101]
						Grain Zn	20.9%	
						Height	30.2%	
						Stem diameter	87.5%	
						Plant defense	14%	
	K (75% CNK)	39–79	*Zea mays*	60 days after sowing	Pot experiment	Fresh and dry biomass	47–51%	[102]
						Fresh shoot biomass	8.4-fold	
						Dry shoot biomass	10-fold	
						N uptake	8.4-fold	
						P uptake	11.4-fold	
	Urea (100%)		*Solanum tuberosum*	90 days	Pot experiment	Fresh weight	95.6%	[103]
						Dry weight	116%	
	CuO- chitosan/alginate NPs	~300	*Fortunella margarita* Swingle		Petri dishes	Germination seed	10%	[104]
Mesoporous silica nanoparticle	-	20	Wheat	6–14 days	Petri dishes (controlled conditions)	Germination rate	12.8%	[105]
						Shoot fresh weight	30.4%	
						Root fresh weight	50%	
						Chlorophyll content	38.4%	
						Total proteins	17.7%	
	Auxin on mesoporous Au/SiO_2_	40–60	*Linum usitatissimum*	3 weeks	Growth chamber (controlled conditions)	Embryogenesis	65%	[106]
						Calli induction frequency	6%	
						Calli length	31.2%	
						Number of regeneration zones	3.6-fold	
	Nanocomposite of ZnAl_2_Si_10_O_24_ + urea	55.2	*Oryza sativa*	14 days	Pot experiment	Nitrogen recovery efficiency	~10%	[107]
Amorphous calcium phosphate	*Glomus mosseae* *Piriformospora indica*	88	*Zea mays*	45 days	Pot experiment	Shoot length	8.3%	[108]
						Root length	17.2%	
						Shoot dry weight	14.6%	
						Shoot fresh weight	39.44%	
						Root fresh weight	54.3%	
	Urea	30–100	Tempranillo grapevine	7 weeks	Field condition	Arginine	~70%	[109]
						Amino N	~21%	
						YAN (N content)	~64%	
	NPK	10–25	*Triticum durum*	-	Pot experiment	Nitrogen efficiency	40%	[110]
						Kernel weight	~73%	
	Urea	13.8	*Triticum durum*	-	Growth chamber Field condition	Plant weight	~40%	[111]
						Ear weight	~60%	
						Ear number	~50%	
						Kernel number	~27%	
	Urea	~10	*Vitis vinifera* L. cv Pinot Gris	Two season of study (2019–2020)	Pot experiment (semi-controlled conditions)	Chlorophyll (SPAD)	~10%	[112]
						Yield	~40%	
						Bunch weight	~46%	
						YAN	~53%	
	Urea	~10	*Cucumis sativus* L	7 days	Hydroponiccondition	Root biomass	~120%	[113]
						Shoot biomass	~25%	
						Root N concentration	~32%	

Abbreviations: FTIR: Fourier transform infrared spectroscopy, DLS: dynamic light scattering, XRD: X-ray diffraction, SEM: scanning electron microscopy, TEM: transmission electron microscopy, XPS: X-ray photoelectron spectroscopy, NTA: nanoparticle tracking analysis, TGA: thermal gravimetric analysis, SEM-EDX: scanning electron microscopy–energy dispersive X-ray spectroscopy, FESEM: field emission scanning electron microscopy, DSC: differential scanning calorimetry, AFM: Atomic force microscopy, NMR: nuclear magnetic resonance, ICP-OES: inductively coupled plasma-optical emission spectroscopy, BET: Brunauer–Emmett–Teller, DTG: differential thermogravimetric.

## Data Availability

Not applicable.

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
