# Peer review of "Eco-Efficient Systems Based on Nanocarriers for the Controlled Release of Fertilizers and Pesticides: Toward Smart Agriculture"

_nanomaterials, 2023, doi:10.3390/nano13131978_

Round 1

Reviewer 1 Report

I suggest to have your article proof-read by a native English speaking chemist before resubmission.

There are a lot of grammatical and language mistakes that should be corrected before re-submission

Author Response

Dear reviewer

We would like to thank all your comments and corrections to our manuscript entitled “Eco-efficient systems based on nanocarriers to the controlled release of fertilizers and pesticides: Toward smart agriculture” (Nanomaterials-2414793) from Fincheira et al.. Considering that, we answer your suggestion below.

Reviewer suggestion

  • I suggest to have your article proof-read by a native English speaking chemist before resubmission.
  • There are a lot of grammatical and language mistakes that should be corrected before re-submission

Author response

  • The manuscript was fully checked using the Grammarly Premium software

*Note:  The changes made to the manuscript with suggested revisions are marked in red.

Looking forward to hearing from you, I remain with best regards,

 Dr. Paola Alejandra Fincheira Robles

Universidad de La Frontera, Casilla 54-D Temuco, Chile

e-mail: p.fincheira01@ufromail.cl

Reviewer 2 Report

Many papers have been presented and reviewed in this paper. These results are classified into 9 types for pesticides and 5 types for fertilisers. Nanomaterials in the agricultural field would be effective in the future. The effectiveness of nanoparticles containing pesticides and fertilizers is well demonstrated and reviewed in this paper.

Author Response

Temuco, June, 2023

Paola Fincheira Robles

Centro de Excelencia en Investigación Aplicada al Medio Ambiente

Universidad de La Frontera

Temuco-Chile

p.fincheira01@ufromail.cl

Dear reviewer

We would like to thank all your comments to our manuscript entitled “Eco-efficient systems based on nanocarriers to the controlled release of fertilizers and pesticides: Toward smart agriculture” (Nanomaterials-2414793) from Fincheira et al.. Considering that, we answer your suggestion below.

Reviewer suggestion

  • Many papers have been presented and reviewed in this paper. These results are classified into 9 types for pesticides and 5 types for fertilisers. Nanomaterials in the agricultural field would be effective in the future. The effectiveness of nanoparticles containing pesticides and fertilizers is well demonstrated and reviewed in this paper.

Author response

The authors appreciate your positive comments and appreciation of our work.

*Note: The changes made to the manuscript with suggested revisions are marked in red.

Looking forward to hearing from you, I remain with best regards,

Dr. Paola Alejandra Fincheira Robles

Departamento de Ingeniería Química

Universidad de La Frontera, Casilla 54-D Temuco, Chile

e-mail: p.fincheira01@ufromail.cl

Reviewer 3 Report

This review manuscript (et al) is compiling and describing recent developments in the applications of nanomaterials in the field of agrochemicals such as pesticides and fertilizers. It aims to focus on how nanomaterials are employed through controlled release systems in addressing environmental, ecological, and health-related issues associated with such agrochemicals. Overall, this manuscript summarizes types of such nanomaterials, their utility for agrochemicals and their future potential. It can serve as a readable summary and its scope fits in a special topic in Nanomaterials. However, despite such broad positive perspectives, there are several areas that it needs to strengthen and improve as commented more in details below. These would need to be addressed prior to its consideration for publication.

Comments:

1.     This manuscript lacks description of design principles of nanomaterial designs that are enabling environmentally responsive controlled release. It is recommended to provide loading principles more specifically and mechanisms engaged in controlled release of loaded agrochemicals.

2.     This manuscript has four figures presented. These are not enough for nanomaterial understanding as well as their quality needs a significant improvement. All of these are depicted in a generic manner and do not provide useful information. In addition, authors are strongly encouraged to prepare figures using original figures copied from their articles with copyrights permission.

Author Response

Temuco, June, 2023

Paola Fincheira Robles

Centro de Excelencia en Investigación Aplicada al Medio Ambiente

Universidad de La Frontera

Temuco-Chile

p.fincheira01@ufromail.cl

Dear reviewer

We would like to thank all your comments to our manuscript entitled “Eco-efficient systems based on nanocarriers to the controlled release of fertilizers and pesticides: Toward smart agriculture” (Nanomaterials-2414793) from Fincheira et al.. Considering that, we answer your suggestions below.

Reviewer suggestion 1

  • This manuscript lacks description of design principles of nanomaterial designs that are enabling environmentally responsive controlled release. It is recommended to provide loading principles more specifically and mechanisms engaged in controlled release of loaded agrochemicals.

Author response to suggestion 1

  • A one paragraph with description of design and mechanism principles of nanocarriers with response to environmental stimulus was added at the beginning of each section that include Responsive to pH, Responsive to temperature, and Responsive to light.

Reviewer suggestion 2

  • This manuscript has four figures presented. These are not enough for nanomaterial understanding as well as their quality needs a significant improvement. All of these are depicted in a generic manner and do not provide useful information. In addition, authors are strongly encouraged to prepare figures using original figures copied from their articles with copyrights permission.

Author response to suggestion 2

  • The quality of figures were improved to PNG 300 DPI. Thank you for your comment. The authors could have figure in their research manuscripts. However, none of these can represent the same information what the figure added in the manuscript shows. The figure added in the text and made by the authors of the review, summarized all the current information related with the review topics treated. 

*Note: The changes made to the manuscript with suggested revisions are marked in red.

Looking forward to hearing from you, I remain with best regards,

Dr. Paola Alejandra Fincheira Robles

Departamento de Ingeniería Química

Universidad de La Frontera, Casilla 54-D Temuco, Chile

e-mail: p.fincheira01@ufromail.cl

Reviewer 4 Report

The review work presented by the authors contains some interesting hypotheses. However, before it can be considered for publication, the work still needs some improvements.

1. Section 2, Controlled release system: A novel nanotechnological approach, is very short. It should be either enhanced, or merged with section 3 (which is also very short), with appropriate section title change.

2. Tables 1-3 - the characterization methods are not of particular importance, in my opinion. However, their effect should be presented and quantified (for Table 1 - observed inhibition, table 2 and 3 - observed effect, etc)

3. The manuscript generally lacks the authors' comments and opinions on the presented data. For each section and sub-section, there is a need for a brief discussion and author's personal opinions. 

Author Response

Temuco, June, 2023

Paola Fincheira Robles

Centro de Excelencia en Investigación Aplicada al Medio Ambiente

Universidad de La Frontera

Temuco-Chile

p.fincheira01@ufromail.cl

Dear reviewer

We would like to thank all your comments to our manuscript entitled “Eco-efficient systems based on nanocarriers to the controlled release of fertilizers and pesticides: Toward smart agriculture” (Nanomaterials-2414793) from Fincheira et al.. Considering that, we answer your suggestions below.

Reviewer suggestion 1 

  • Section 2, Controlled release system: A novel nanotechnological approach, is very short. It should be either enhanced, or merged with section 3 (which is also very short), with appropriate section title change.

Author response to suggestion 1 

  • The authors merged sections 2 and 3, and created section 2 with the title “A smart agricultural technology based on controlled release system”

Reviewer suggestion 2 

  • Tables 1-3 - the characterization methods are not of particular importance, in my opinion. However, their effect should be presented and quantified (for Table 1 - observed inhibition, table 2 and 3 - observed effect, etc).

 Author response to suggestion 2 

  • The data presented in table 1-3 were revised and quantified to present in each reference.

Reviewer suggestion 3 

  • The manuscript generally lacks the authors' comments and opinions on the presented data. For each section and sub-section, there is a need for a brief discussion and author's personal opinions.

Author response to suggestion 3

  • Comment or opinion recommended for the reviewer were added to each section.

*Note: The changes made to the manuscript with suggested revisions are marked in red.

Looking forward to hearing from you, I remain with best regards,

Dr. Paola Alejandra Fincheira Robles

Departamento de Ingeniería Química

Universidad de La Frontera, Casilla 54-D Temuco, Chile

e-mail: p.fincheira01@ufromail.cl

Round 2

Reviewer 1 Report

Line 20 ...on t human health. Change to on human health (delete t).

Line 53 ...harm on human health ... delete on.

Line 55 ... they can have damage in animals...Change to: they can damage animals...

Line 69 Change to: It has identified promising...  

Lin3 70 delete t

The sentence in Lines 212-214 needs rephrasing

Line 222 Change to: Nanofibers are considered attractive one-dimensional nanostructures due to...

Line 243 Insert space in essential tool  

Line 273 ...was studied by [69] Give the name of the main Author of Ref 69. ...was studied by Xxxxx et al. [69]

Line 280 delete - 3-mercaptopropyl

Line 360 Change uptake to uptaken

Line 368 Change regarding to compare to

Line 395 Give the full name of the acronym NPK

Line 399 Give the name of the main author (as for line 273)

The same in Line 407

Line 425 Insert space g-1 hydrotalcite

Line 442 Insert space in maintaining

Line 50 Delete extra space to  reduce

Line 458 Delete the second via

Line 485 Give the name of the main author (as for line 273)

Line 498 Change to: ...significant results...

Line 502 Change to: In general, MSN present interesting results, but testing their effects...

Line 517 Give the name of the main author (as for line 273)

Line 526 Give the full name of the acronym SPAD

Line 584-5 Change so far to further 

The English language has significantly improved from the previously submitted version of the manuscript.

Author Response

Temuco, June, 2023

Dear reviewer

We would like to thank all your comments and corrections to our manuscript entitled “Eco-efficient systems based on nanocarriers to the controlled release of fertilizers and pesticides: Toward smart agriculture” (Nanomaterials-2414793) from Fincheira et al.. Considering that, we answer your suggestion below.

  • Reviewer suggestions

Line 20 ...on t human health. Change to on human health (delete t).

Line 53 ...harm on human health ... delete on.

Line 55 ... they can have damage in animals...Change to: they can damage animals...

Line 69 Change to: It has identified promising...  

Lin3 70 delete t

The sentence in Lines 212-214 needs rephrasing

Line 222 Change to: Nanofibers are considered attractive one-dimensional nanostructures due to...

Line 243 Insert space in essential tool  

Line 273 ...was studied by [69] Give the name of the main Author of Ref 69. ...was studied by Xxxxx et al. [69]

Line 280 delete - 3-mercaptopropyl

Line 360 Change uptake to uptaken

Line 368 Change regarding to compare to

Line 395 Give the full name of the acronym NPK

Line 399 Give the name of the main author (as for line 273)

The same in Line 407

Line 425 Insert space g-1 hydrotalcite

Line 442 Insert space in maintaining

Line 50 Delete extra space to  reduce

Line 458 Delete the second via

Line 485 Give the name of the main author (as for line 273)

Line 498 Change to: ...significant results...

Line 502 Change to: In general, MSN present interesting results, but testing their effects...

Line 517 Give the name of the main author (as for line 273)

Line 526 Give the full name of the acronym SPAD

Line 584-5 Change so far to further 

  • Author response

The authors are grateful for the English corrections made by the reviewer. All English grammar suggestions were corrected

*Note:  The changes made to the manuscript with suggested revisions are marked in red.

Looking forward to hearing from you, I remain with best regards,

 Dr. Paola Alejandra Fincheira Robles

Universidad de La Frontera, Casilla 54-D Temuco, Chile

e-mail: p.fincheira01@ufromail.cl

Reviewer 3 Report

Reviewers’ comments are made addressed in part by describing design principles of controlled release and examples of pay loading. However, despite reviewer's comments, previous  figures appear almost unchanged, and fail to offer more impactful images and attract readability.

Page 2 (line 72). “Specifically, nanoparticles (NPs) used as a nanocarrier system are particles with a size between 50 and 1000 nm,” Examples summarized in Tables are showing some nanoparticles of less than 20 nm. Please clarify the source of 50 nm.

Some minor errors and typos are found throughout the manuscript that include: “on t human health (page 1, line 20, abstract), an inconsistent reference citation (page 8, line 406), “inmaintaining ..” (page 9, line 442). These are minors that can be fixed during galley proofing.

Author Response

Temuco, June, 2023

Dear reviewer

We would like to thank all your comments and corrections to our manuscript entitled “Eco-efficient systems based on nanocarriers to the controlled release of fertilizers and pesticides: Toward smart agriculture” (Nanomaterials-2414793) from Fincheira et al.. Considering that, we answer your suggestion below.

  • Reviewer suggestion 1

Reviewers’ comments are made addressed in part by describing design principles of controlled release and examples of pay loading. However, despite reviewer's comments, previous figures appear almost unchanged, and fail to offer more impactful images and attract readability.

  • Author response 1

The figures were modified as requested by the reviewer

 Reviewer suggestion 2

 Page 2 (line 72). “Specifically, nanoparticles (NPs) used as a nanocarrier system are articles with a size between 50 and 1000 nm,” Examples summarized in Tables are showing some nanoparticles of less than 20 nm. Please clarify the source of 50 nm.

  • Author response 2

The authors appreciate the annotation made by the author. Therefore, the description of the concept was modified along the same lines.

  • Reviewer suggestion 3

 Some minor errors and typos are found throughout the manuscript that include: “on t human health (page 1, line 20, abstract), an inconsistent reference citation (page 8, line 406), inmaintaining ..” (page 9, line 442). These are minors that can be fixed during galley proofing.

  • Author response 3

The authors appreciate the language suggestions, which were corrected.

*Note:  The changes made to the manuscript with suggested revisions are marked in red.

Looking forward to hearing from you, I remain with best regards,

 Dr. Paola Alejandra Fincheira Robles

Universidad de La Frontera, Casilla 54-D Temuco, Chile

e-mail: p.fincheira01@ufromail.cl
